# Plasma Galectin-4 Levels Are Increased after Stroke in Mice and Humans

**DOI:** 10.3390/ijms241210064

**Published:** 2023-06-13

**Authors:** Amra Jujic, João P. P. Vieira, Hana Matuskova, Peter M. Nilsson, Ulf Lindblad, Michael H. Olsen, João M. N. Duarte, Anja Meissner, Martin Magnusson

**Affiliations:** 1Wallenberg Centre for Molecular Medicine, Lund University, 22100 Lund, Sweden; 2Department of Clinical Sciences, Lund University, 20502 Malmö, Sweden; 3Department of Cardiology, Skåne University Hospital, 21428 Malmö, Sweden; 4Department of Experimental Medical Science, Lund University, 22100 Lund, Sweden; 5German Center for Neurodegenerative Diseases, 53127 Bonn, Germany; 6General Practice—Family Medicine, School of Public Health and Community Medicine, Institute of Medicine, Sahlgrenska Academy, University of Gothenburg, 40530 Gothenburg, Sweden; 7Department of Internal Medicine 1, Holbaek Hospital, 4300 Holbaek, Denmark; 8Department of Regional Health Research, University of Southern Denmark, 5000 Odense, Denmark; 9Department of Physiology, Institute for Theoretical Medicine, University of Augsburg, 86159 Augsburg, Germany; 10Hypertension in Africa Research Team (HART), North-West University, Potchefstroom 2520, South Africa

**Keywords:** ischemic stroke, cardiovascular disease, diabetes, obesity, galectins

## Abstract

Epidemiological studies have associated plasma galectin-4 (Gal-4) levels with prevalent and incident diabetes, and with an increased risk of coronary artery disease. To date, data regarding possible associations between plasma Gal-4 and stroke are lacking. Using linear and logistic regression analyses, we tested Gal-4 association with prevalent stroke in a population-based cohort. Additionally, in mice fed a high-fat diet (HFD), we investigated whether plasma Gal-4 increases in response to ischemic stroke. Plasma Gal-4 was higher in subjects with prevalent ischemic stroke, and was associated with prevalent ischemic stroke (odds ratio 1.52; 95% confidence interval 1.01–2.30; *p* = 0.048) adjusted for age, sex, and covariates of cardiometabolic health. Plasma Gal-4 increased after experimental stroke in both controls and HFD-fed mice. HFD exposure was devoid of impact on Gal-4 levels. This study demonstrates higher plasma Gal-4 levels in both experimental stroke and in humans that experienced ischemic stroke.

## 1. Introduction

In previous epidemiological studies, we have shown that increased levels of galectin-4 (Gal-4) are linked with prevalent and incident diabetes [1], as well as increased risk of future myocardial infarction, heart failure, cardiovascular, and all-cause mortality [2]. Whether plasma Gal-4 is also associated with ischemic stroke, which is a major complication in diabetic patients, has not yet been investigated.

Gal-4 is a gastrointestinal tract protein involved in the apical trafficking of proteins, including dipeptidyl peptidase-4 (DPP-4), which accumulates intracellularly in Gal-4-depleted mice [3]. Soluble and membrane-bound DPP-4 inactivate gastric inhibitory polypeptide (GIP) and glucagon-like peptide 1 (GLP-1), which are largely responsible for the incretin effect [4]. The latter is defective in individuals with diabetes [5], leading to cardio-metabolically adverse effects [6]. Recent evidence suggests that incretin-based antihyperglycemic agents, including DPP-4 and GLP-1 receptor agonists, exert beneficial effects in patients with diabetes who suffer ischemic stroke by reducing infarct size and promoting recovery [7,8,9,10].

Here, we tested whether plasma Gal-4 concentrations increase after stroke and whether this may be dependent on metabolic syndrome by using a mouse model of ischemic stroke. The clinical relevance of potential stroke-associated Gal-4 alterations was assessed by testing association with prevalent stroke in a population-based cohort study.

## 2. Results

### 2.1. Gal-4 Plasma Levels Increase after Experimental Ischemic Stroke

To induce metabolic syndrome, mice were fed a high-fat diet (HFD) for 8 weeks before stroke induction (Figure 1a). Compared to the control diet (CD; 10% fat kcal), HFD (60% fat kcal) over the course of 8 weeks led to increased body weight (in g: 29.8 ± 0.9 in CD vs. 40.5 ± 1.2 in HFD, *p* < 0.001), higher plasma glucose (in mmol/L: 4.4 ± 0.3 in CD vs. 6.2 ± 0.5 in HFD, *p* = 0.007), and elevated plasma insulin (in µg/L: 3.2 ± 1.8 in CD vs. 15.8 ± 5.7 in HFD, *p* = 0.035).

Stroke induced by transient middle cerebral artery occlusion (tMCAo) was confirmed by lower neurological function (i.e., higher neuro-scores after induction of stroke compared to sham surgery: 0 in CD and HFD for sham vs. 8.8 ± 5.2 in CD and 6.7 ± 3.2 in HFD for tMCAo) and apparent ischemic lesions (41 ± 13% in CD, 37 ± 15% in HFD). Mice on CD and HFD had a poorer survival rate after tMCAo compared to mice on conventional rodent chow (survival at 1 day after tMCAo in %: 100 for conventional rodent chow vs. 83 for CD and 50 for HFD).

While HFD had negligible impact on plasma Gal-4, tMCAO resulted in increased Gal-4 plasma levels, as measured 3 days after stroke (Figure 1b; ANOVA effects: diet F(1,11) = 5.00, *p* = 0.047, stroke F(1,11) = 15.50, *p* = 0.002, interaction F(1,11) = 0.081, *p* = 0.781).

An independent mouse cohort confirmed the stroke-associated increase of plasma Gal-4 concentrations (Figure 2). Plasma Gal-4 concentration was elevated in mice fed conventional rodent chow (5.3% fat kcal) at 1 day post-stroke compared to sham-operated controls (*p* = 0.0129). Moreover, plasma Gal-4 levels did not differ between sham-operated mice on the conventional chow diet, CD or HFD (Gal-4 concentration in µg/µL: 3.4 ± 0.6 for rodent chow, 3.8 ± 0.7 for CD, 5.6 ± 0.4 for HFD; ANOVA: *p* = 0.1283).

### 2.2. Gal-4 Associates with Prevalent Stroke

Subjects with prevalent ischemic stroke (n = 59) were older, more often men, had diabetes mellitus to higher extent, and were more often treated for hypertension. Gal-4 was significantly higher in subjects with prevalent ischemic stroke than in subjects without (Table 1).

In logistic regressions, Gal-4 was associated with prevalent ischemic stroke in an unadjusted model and following adjustments for age and sex as well as the main covariates related to weight, diabetes, and cardiovascular health (Table 2). In univariate regression analyses, each doubling of Gal-4 concentration was associated with higher BMI (β 0.87; *p* = 8.3 × 10^−8^) and a higher probability of antihypertensive treatment (OR 2.24 [1.90–2.63]; *p* = 2.2 × 10^−22^). Gal-4 had no significant association with systolic blood pressure (*p* = 0.678).

## 3. Discussion

This is the first study to show higher plasma Gal-4 levels in patients with prevalent stroke. Verification in a murine model of experimental stroke supports a direct interaction between Gal-4 plasma levels and ischemic stroke. The association of increases in plasma Gal-4 with higher BMI in the tested human cohort necessitated the investigation of whether HFD feeding has an effect on Gal-4 levels in a controlled experimental setting. Testing whether HFD feeding increases Gal-4 plasma levels through the assessment of Gal-4 levels after 8 weeks of HFD prior to stroke revealed that HFD has a negligible impact on plasma Gal-4 levels in mice. From these data, it can be concluded that impaired fasting glucose and diabetes status (both apparent in mice after 8 weeks of HDF exposure) are not directly linked to alterations in Gal-4 levels. Although this is in contrast to results from previously published epidemiological studies that report associations between high Gal-4 levels and prevalent as well as incident diabetes [1], our findings showing a direct effect of stroke on plasma Gal-4 elevation are supported by studies reporting similar relationships between plasma Gal-4 and the incidence of myocardial infarction [2], heart failure [2,11], and cardiovascular and all-cause mortality [2]. Together, these data rather suggest that changes in Gal-4 levels are generally associated with the occurrence of cardiovascular events, including stroke. This notion is underpinned by the results of the current investigation, showing that stroke-induced plasma Gal-4 elevation is independent of metabolic syndrome.

The mechanisms by which Gal-4 concentration increase post-stroke are not yet known. Nonetheless, several pathogenic processes post-stroke may be affected by altered Gal-4 homeostasis. Gal-4 has been reported to be involved in immunoregulatory functions through the activation and differentiation of monocytes [12], which have been proposed as central players in the detrimental innate proinflammatory response post-stroke [13]. In addition to inflammation, the formation of new vessels through angiogenesis is thought to participate in functional stroke recovery [14]. Gal-4 has been linked to the augmented secretion of circulating cytokines responsible for endothelial activation related to angiogenesis and thus cancer metastasis [15]. Earlier studies supported its role in the proliferation and migration of different cell types [16], emphasizing a potential involvement in angiogenic processes that may also occur post-stroke. Similarly, arteriogenesis, as a major process to improve collateral flow, has been associated with Gal-4 [17]. As patients with good collateral flow experience more favorable stroke outcomes and metabolic risk factors such as diabetes are associated with poor leptomeningeal collateral status [18], investigating the role of Gal-4 in this respect may be promising.

Taken together, the relatively sparse knowledge base available on the role of Gal-4 in stroke pathology and related processes, as well as its relationship to risk factors for stroke, such as metabolic syndrome, and diabetes, necessitate mechanistic and clinical studies to investigate the relative potential of Gal-4 as a prognostic marker for stroke outcome or even its potential as therapeutic target.

### Study Limitations

The assessed population cohort (i.e., the Malmö Preventive Project Re-Examination cohort) consists of mainly elderly, white European men. Thus, our findings might not be generalizable to the general population. As is common to all observational studies, no conclusions about causality can be drawn. Further, this study did not aim at identifying molecular mechanisms involved in Gal-4 alterations post-stroke. With a fast dissemination of the findings that Gal-4 plasma levels increase with stroke, we instead want to stimulate further research on this important topic and pave the way for additional mechanistic studies that may include the testing of the relative potential of Gal-4 as a prognostic marker for stroke outcome, its potential as a therapeutic target, or the investigation of plasma Gal-4 as prognostic parameter for stroke incidence in patients with diabetes or metabolic syndrome.

## 4. Materials and Methods

### 4.1. Mouse Study

Male C57BL/6J mice (8 weeks old; Taconic, Skensved, Denmark) were group-housed under a 12 h light–dark cycle in enriched cages with access to food and water ad libitum. After acclimatization, at 9 weeks of age, mice were started on high-fat (HFD, 60% fat kcal; N = 12) or control (CD, 10% fat kcal; N = 12) diets (Research Diets, New Brunswick, NJ-USA) for 8 consecutive weeks as previously described [19]. Where possible, experimenters were blinded to group allocation during sample processing and data analysis.

#### 4.1.1. Mouse Model of Stroke

Transient middle cerebral artery occlusion (tMCAo) was performed as previously described [20]. Briefly, the middle cerebral artery was transiently occluded using a monofilament (9–10 mm coating length, 0.19 ± 0.01 mm tip diameter; Doccol, Sharon, MA, USA) in anesthetized mice (isoflurane in 70% N_2_O, 30% O_2_). Reperfusion was initiated after 60 min. Cerebral blood flow was monitored using laser Doppler flowmetry (Moor Instruments, Axminster, UK). The same protocol without occlusion was used for sham surgery. Neurological function was evaluated using the sum of focal and general scores ranging between 0 (no deficits) and 56 (the poorest performance) [20]. Coronal brain slices (1 mm thick) were stained with 2,3,5-triphenyltetrazolium chloride (Sigma-Aldrich, #93140), and infarct area was determined and presented as percentage of contralateral hemisphere [20]. Group sizes were determined based on prior experiments, considering estimated group specific attrition. Mice from CD (n = 12) and HFD (n = 12) groups were randomly assigned to tMCAo or sham surgery. In the CD group, 6 mice were subjected to tMCAo and 6 mice to sham surgery, while 8 of the HFD mice were subjected to tMCAo and 4 to sham surgery. A separate cohort of mice fed on conventional rodent chow (SAFE A30; Safe Diets, Augy, France) was utilized to assess direct effects of ischemic stroke on plasma Gal-4 levels (n = 8 for sham and tMCAo, respectively).

#### 4.1.2. Plasma Analyses

Pre-surgery blood from the saphenous vein and post-stroke blood withdrawn from vena cava prior to transcardial perfusion was collected into EDTA-coated tubes (#41.1395.105, Sarstedt, Nürnbrecht, Germany), and separated by centrifugation at 1000× *g* for 10 min at room temperature. Plasma glucose was measured using the glucose oxidase method coupled to a peroxidase reaction oxidizing the AmplexRed reagent (#A12222, Invitrogen, Fisher Scientific, Göteborg, Sweden), as detailed before [21]. Commercially available ELISA kits were used to determine plasma insulin (#10-1247-10, Mercodia, Uppsala, Sweden) and Gal-4 (#NBP2-76725, Novus Biologicals, Bio-Techne, Abingdon, UK).

#### 4.1.3. Statistics

Data were analyzed by Prism 9.3.0 (GraphPad, San Diego, CA, USA). After normality testing (Kolmogorov–Smirnov and Shapiro–Wilk tests), data were analyzed by either Student’s *t*-test (HFD effect pre-stroke) or Mann–Whitney (stroke effect) or 2-way ANOVAs with diet and stroke as factors. Significant diet, stroke, or interaction effects were followed by Fisher’s least significant difference (LSD) tests for independent comparisons. Normally distributed data are presented as mean ± standard error of the mean (SEM). Data that are not normally distributed are presented as median ± interquartile range (IQR).

### 4.2. Human Study

Within the Malmö Preventive Project [22] Re-Examination cohort, a sub-sample of participants was randomly selected based on glucometabolic status [23], i.e., 1/3 normoglycemic; 1/3 with impaired fasting glucose (IFG), and 1/3 with diabetes (n_total_ = 1792). Blood samples provided by 1737 individuals were analyzed with proximity extension assay technology (Proseek Multiplex CVD III from Olink Bioscience, Uppsala, Sweden). The CVD III panel consists of 92 proteins, among them galectin-4, with either established or proposed associations with CVD, inflammation, and metabolism. Complete data for all co-variates were available in 1688 subjects.

#### 4.2.1. Examinations

Standardized methods were used to measure anthropometrics and blood pressure, as described elsewhere [23]. Fasting plasma glucose and high-density lipoprotein (HDL) cholesterol were analyzed using a Beckman Coulter LX20 (Beckman Coulter, Brea, CA, USA) at Department of Chemistry, Skåne University Hospital, Malmö. Fasting blood samples for proteomic analyses were stored at −80 °C until time of analysis.

#### 4.2.2. Definitions

Prevalent stroke was defined as prior ischemic stroke, and data were collected through regional and national registers, defined as ICD9 codes 433–434 or ICD10 codes I63.0–I63.9. IFG was defined as fasting plasma glucose ≥5.6 mmol/L. Prevalent diabetes was defined as either previously known diabetes or new-onset diabetes (two separate measurements of fasting plasma glucose ≥7.0 mmol/L or one measurement ≥11.1 mmol/L) [23]. Smoking was self-reported and defined as present smoking. Antihypertensive treatment was defined as use of any blood pressure-lowering medicine and retrieved through the National Prescribed Drug Register (starting in 2005).

#### 4.2.3. Statistics

Associations between Gal-4 and prevalent ischemic stroke were explored using logistic regression models in three steps: (1) unadjusted; (2) adjusted for age and sex (Model (1), and (3) adjusted for body mass index, systolic blood pressure, prevalent IFG/diabetes, HDL-cholesterol, antihypertensive treatment, and smoking (Model 2). Groups were compared using Student’s *t*-tests or χ^2^ tests, where appropriate. Associations between Gal-4 and continuous variables were analyzed using univariate linear regressions, and associations between Gal-4 and binary variables using univariate logistic regressions.

## Figures and Tables

**Figure 1 ijms-24-10064-f001:**
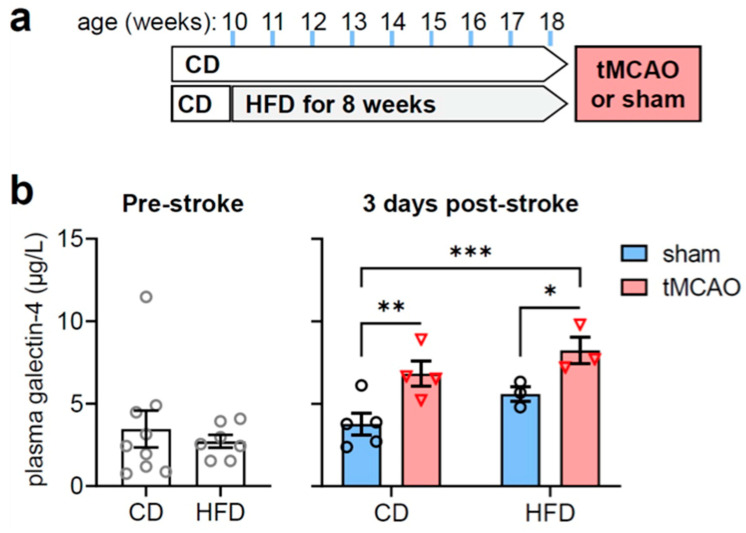
Stroke-associated increases in galectin-4 plasma concentrations are independent of metabolic syndrome. (**a**) Study design: mice were fed HFD or CD for 8 consecutive weeks. (**b**) Plasma galectin-4 levels after 8 weeks of HFD or CD feeding (left panel), and 3 days after stroke or sham surgery in mice fed the HFD and CD (right panel). Data are shown as mean ± SEM. Symbols (open grey circles = pre-stroke, open black circles = post-sham surgery, open red triangles = post-tMCAo) represent individual mice. * *p* < 0.05, ** *p* < 0.01, *** *p* < 0.001 based on Fischer’s LSD post-hoc comparison after significant effect of diet and stroke in ANOVA. CD = control diet, HFD = high-fat diet, tMCAo = transient middle cerebral artery occlusion.

**Figure 2 ijms-24-10064-f002:**
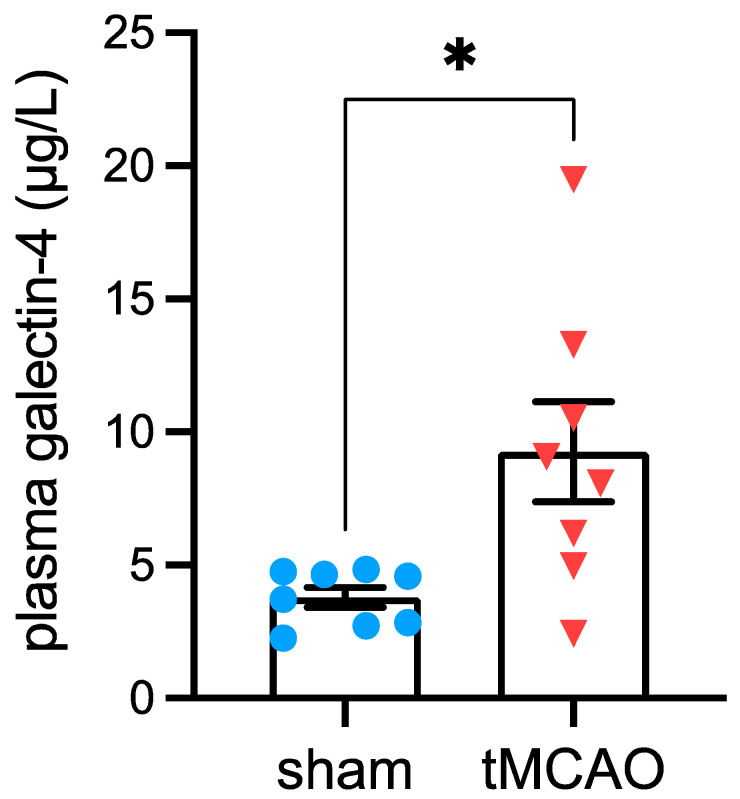
Stroke-associated increases in galectin-4 plasma concentrations. Plasma galectin-4 levels 1 day after tMCAo or sham surgery in mice fed conventional rodent chow. Data are shown as mean ± SEM. Symbols (full blue circles = post-sham surgery, full red triangles = post-tMCAo) represent individual mice. * *p* < 0.05 based unpaired t-test. tMCAo = transient middle cerebral artery occlusion.

**Table 1 ijms-24-10064-t001:** Characteristics of the human study population. Values are means (±standard deviation) and numbers (percent).

Characteristics of the Study Population				
	Total	Subjects Free from Ischemic Stroke	Subjects with Prevalent Ischemic Stroke	*p*
	n = 1688	n = 1629	n = 59	
**Age (years)**	67.4 (±6.0)	67.3 (±6.0)	70 (±5.4)	3.0 × 10^−4^
**Sex (women; n (%))**	491 (29.1)	482 (29.6)	9 (15.3)	0.017
**BMI (kg/m^2^)**	28.3 (±4.3)	28.4 (±4.4)	28.0 (±3.8)	0.411
**IFG (n (%)) ***	627 (37.1)	610 (37.5)	17 (28.8)	0.175
**Diabetes mellitus (n (%))**	679 (40.2)	646 (39.7)	33 (55.9)	0.012
**SBP (mmHg)**	147 (±20)	147 (±20)	148 (±20)	0.683
**HDL-cholesterol (mmol/L)**	1.3 (±0.4)	1.3 (±0.4)	1.3 (±0.4)	0.902
**Smoking (n (%))**	306 (18.1)	294 (18.0)	12 (20.3)	0.654
**AHT (n (%))**	788 (46.7)	740 (45.4)	48 (81.4)	5.5 × 10^−8^
**Gal-4 (AU)**	3.1 (±0.7)	3.1 (±0.6)	3.4 (±0.6)	6.8 × 10^−4^

AHT = antihypertensive treatment; AU = arbitrary unit (NPX in log_2_ scale where 1 NPX increase means a doubling of protein concentration); BMI = body mass index; Gal-4 = galectin-4; HDL = high-density lipoprotein; IFG = impaired fasting glucose; NPX = normalized protein expression; SBP = systolic blood pressure. * Fasting plasma glucose ≥5.6 mmol/L.

**Table 2 ijms-24-10064-t002:** Associations between plasma galectin-4 and prevalent stroke.

Logistic Regression Analyses		
	OR (CI 95%)	*p*
**UNADJUSTED**		
Galectin-4	1.99 (1.35–2.92)	4.7 × 10^−4^
**MODEL 1**		
Galectin-4	1.74 (1.17–2.59)	0.006
Age	1.08 (1.03–1.14)	0.001
Sex	0.33 (0.16–0.69)	0.003
**MODEL 2**		
Galectin-4	1.52 (1.01–2.30)	0.048
Age	1.06 (1.01–1.11)	0.021
Sex	0.34 (0.16–0.73)	0.006
Body mass index	0.95 (0.88–1.02)	0.145
IFG/diabetes mellitus *	1.18 (0.55–2.57)	0.669
HDL-cholesterol	1.36 (0.66–2.80)	0.408
Systolic blood pressure	1.00 (0.99–1.01)	0.792
Smoking	1.34 (0.68–2.64)	0.399
Antihypertensive treatment	4.49 (2.23–9.05)	2.6 × 10^−5^

Values are odds ratios (OR) with 95% confidence intervals (CI 95%). BMI = body mass index; HDL = high-density lipoprotein; IFG = impaired fasting glucose. * Fasting plasma glucose ≥5.6 mmol/L.

## Data Availability

The data that support the findings of this study are available upon request from Steering Committee of the Malmö Preventive Project study by contacting data manager Anders Dahlin (anders.dahlin@med.lu.se), but restrictions apply to the availability of these data, which were used under license for the current study, and so are not publicly available due to ethical and legal restrictions related to the Swedish Biobanks in Medical Care Act (2002:297) and the Personal Data Act (1998:204).

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
