# Peer review of "Plasma Galectin-4 Levels Are Increased after Stroke in Mice and Humans"

_ijms, 2023, doi:10.3390/ijms241210064_

Round 1
Reviewer 1 Report
I appreciate the opportunity to review the manuscript “Plasma galectin-4 levels are increased after stroke in mice and 2 humans” submitted in the journal International Journal of Molecular Sciences.
The authors investigated the possible role of plasma galectin - 4 in ischemic stroke in animal models and its relationship with the traditional risk factors for ischemic stroke.
Reviewer Comments:
Major recommendations
1. Please extend the introduction and discussion section with the concrete results of the epidemiological and molecular studies.
Minor recommendations
1. Please prepare a list of abbreviations used in manuscript.
2. Please insert the materials and methods before the results and discussion.
Considering that the manuscript gives sufficient new information about the possible role of the plasma galectin - 4 in ischemic stroke in animal models, I think this submission meets the criteria to be published in journal International Journal of Molecular Sciences after the major revision I suggested.
Author Response
Dear Editor
Dear Reviewer,
Thank you for taking the time to review our work and for the opportunity to submit a revised version of the manuscript. We would like to thank the editor and the reviewers for their insightful comments, as we believe that these comments have led to a substantial improvement of the manuscript. We have revised the manuscript according to the editor’s and reviewers’ suggestions, and we hope that it has improved to the level of satisfaction. Detailed responses to reviewers’ comments are given in the attached document. All changes are marked “red” in the revised manuscript.
Yours sincerely,
Corresponding author
Anja Meissner, PhD
On behalf of all the co-authors

Reviewer 2 Report
The authors examine galectin-4 leveles in stroke survivors as well as controls an furthermore in a rodent stroke model.
There are some points the authors should address:
1. There are several orthographical and syntax errors throughout the manuscript.
2. What does prevalent stroke mean? Previous stroke?
3. Please indicate the meaning of abbreviation when used for the first time (e.g., CD, line 56)
4. The authors want to demonstrate an association or relationship between experimental ischemic stroke as well as the Gal-4 level and have shown that Gal-4 increases after stroke. However, the presented experiment does neither demonstrate nor prove an association between stroke and Gal-4 as a risk factor (or even cause) of stroke, since the experimental setting as presented in this study can’t answer this important research question. To address the question whether Gal-4 leads more often to an ischemic stroke, the authors should have used a stroke prone rodent model (e.g. spontaneous hypertensive stroke prone rats (SHR-SP)), which had to be divided in those fed with HFD and in CD. If the HFD-SHR-SP would show more spontaneous strokes, then, a causal relationship may be assumed between stroke and Gal-4. Why an elevation of Gal-4 after stroke is important for the stroke pathogenesis or functional outcome is neither demonstrated nor discussed.
5. How many animals did the authors use for the control and the treatment group? How many animals have died in each experimental group?
6. Fig 1b: Ischemic stroke results in an increase of Gal-4 in mice regardless of whether fed with HFD or with control food. Is there a statistical difference between column 2 and 4, right panel, Fig 1b??
It seems to me that an elevation of Gal-4 is rather a consequence of tMCAo and thus, Gal-4 does not contribute to the risk of stroke. An explanation why Gal-4 after stroke increase is lacking. The elevation of Gal-4 after stroke is perhaps due to the fact that mice after tMCAo are too weak to eat sufficiently. In conclusion, it is not clear what the authors really want to show when performing the animal experiment as presented in the manuscript.
7. The group of patients without stroke and that of stroke survivors are not balanced regarding the group size and thus meaningful statistics or statements are not possible in this setting.
8. What is an “arbitrary unit”? Please clarify.
9. Gal-4 (AU)-levels between “stroke-free” patients (3.1) and stroke survivors (3.4) are similar, however, the p-value is highly significant. I wonder whether the different group size may bias the presented result.
Author Response

(The authors gave the same response as above.)

Round 2
Reviewer 1 Report
I agree that manuscript meets criteria to be published after revision I suggested.
Reviewer 2 Report
The authors have carefully addressed all the points I raised and changed the manuscript accordingly. Over all, the paper has improved much. I have no further concerns.